# Targeting CDK4/6 for Anticancer Therapy

**DOI:** 10.3390/biomedicines10030685

**Published:** 2022-03-16

**Authors:** Jiating Qi, Zhuqing Ouyang

**Affiliations:** 1The Second Clinical College, Tongji Medical College, Huazhong University of Science and Technology, Wuhan 430030, China; jiatingqi@hust.edu.cn; 2Department of Pathogen Biology, School of Basic Medicine, Tongji Medical College, Huazhong University of Science and Technology, Wuhan 430030, China

**Keywords:** CDK4/6, PROTAC, small molecular inhibitor, drug resistance, cancer, palbociclib, ribociclib, abemaciclib, trilaciclib

## Abstract

Cyclin-dependent kinase 4/6 (CDK4/6) are key regulators of the cell cycle and are deemed as critical therapeutic targets of multiple cancers. Various approaches have been applied to silence CDK4/6 at different levels, i.e., CRISPR to knock out at the DNA level, siRNA to inhibit translation, and drugs that target the protein of interest. Here we summarize the current status in this field, highlighting the mechanisms of small molecular inhibitors treatment and drug resistance. We describe approaches to combat drug resistance, including combination therapy and PROTACs drugs that degrade the kinases. Finally, critical issues and perspectives in the field are outlined.

## 1. Introduction

Cell division is one of the fundamental biological processes, which functions in various physical and pathological activities [1]. The series of stages sequentially occurring in cell division compose a cell cycle, which includes two successive periods: interphase and mitosis. The former features DNA synthesis in the S phase, before and after which there are G1 and G2 phases, preparing for DNA replication and mitosis, respectively. The latter is marked by sister chromatid segregation. Independent of a cell cycle also exists a dormant G0 phase, where most non-proliferative cells in the human body stay [2].

Normally, the cell cycle is monitored by a quality control system called checkpoints, which consist of G1/S checkpoint (DNA damage checkpoint), G2/M checkpoint, and mitotic spindle checkpoint. These checkpoints detect abnormal cell division and then induce cell cycle arrest, allowing cells to repair defects, thus ensuring proper DNA synthesis and chromosome separation. In addition, whether a cell steps into a cell cycle relies on both extrinsic (e.g., growth factors) and intrinsic (e.g., protein synthesis) signals. The lack of these factors leads cells to enter the G0 phase [1]. The majority of human cells are quiescent, except those in the hematopoietic system or gut epithelium [3].

While in malignant cells, the cell cycle is deregulated, which is characterized by abnormal and uncontrollable cell division. Cancer-related cell cycle defects occur via mutations on multiple proteins essential at different stages of the cell cycle [2]. These genomic dysfunctions dispose of cells to acquire more mutations and numerical aberrations in chromosomes, which reflects abnormal cell division; the accumulated genome mutations result in constitutive mitogenic signals and deficiency of response to anti-mitogenic signals, which brings about unscheduled cell division [4].

The cell cycle has to be tightly regulated owing to its critical role in carcinogenesis. In general, transition through the cell cycle is an orderly process regulated by a series of proteins, of which cyclin-dependent kinases (CDKs) are the most critical ones. CDKs are a family of serine/threonine protein kinases, which form heterodimers with their respective regulatory cyclin subunits [5]. CDKs are generally divided into three groups according to their functions: mitosis-related CDKs (CDK1, CDK2, CDK4, and CDK6), which directly promote cell cycle progression, although there are also other CDKs that work in mammalian cell cycle regulation [4]; transcription-related CDKs (CDK7, CDK8, and CDK9); and atypical CDKs (CDK5, CDK14, CDK15, CDK16, CDK17, and CDK18) [1].

In the cell cycle, CDKs are periodically activated at specific points by cyclins [2]. First, in G1, proliferative signals are sensed by D-type cyclins, which activate CDK4 and CDK6. Subsequently, the expression of E-type cyclins activates CDK2, and the formation of CDK2-cyclin E complex is necessary for G1/S progression. Then in the late S phase, CDK2 is activated by cyclin A, driving transition toward G2. At the end of G2, CDK1-cyclin A complex onsets mitosis. Finally, following nuclear envelope breakdown in prophase, A-type cyclins are degraded, and CDK1-cyclin B promotes cells through mitosis [4]. Besides cyclins, CDKs activity is also regulated by CAK (CDK-activating kinase, i.e., CDK7-cyclin H complex) and CKI (CDK inhibitors), which comprise INK4 (inhibitor of CDK4) proteins and Cip/Kip (CDK-interacting protein/kinase inhibitor protein). Under cell cycle defects, checkpoints can be activated via regulation of CDKs activity and therefore prevent daughter cells from inheriting the faults.

Among various CDKs, CDK4 and CDK6 (CDK4/6) are critical because they play a fundamental role in the G1/S transition. CDKs function via phosphorylating specific substrates [1]. In the G1 phase, the most important substrate for CDK4/6 is retinoblastoma susceptibility protein (Rb), which interacts with the E2F transcriptional family in its hypophosphorylated state, whereby it suppresses transcription of target genes. When a cell senses mitogenic signals, CDK4/6 are activated by cyclin D and then phosphorylate Rb, thus relieving the inhibition [5]. CDK4/6 also contributes to this process by separating Cip/Kip proteins from cyclin E-CDK2 complex, which facilitates CDK2 activation and the following complete phosphorylation of Rb. Sequential phosphorylation of Rb by CDK4/6 and CDK2 results in E2F activation and transcription initiation of genes required for S phase progression (Figure 1). On the other hand, CDK4/6 do not sequester Cip/Kip family under antiproliferative signals; thus, cyclin E-CDK2 is inhibited and cell cycle is arrested at G1 phase.

## 2. CDK4/6 Are Attractive Targets for Anticancer Treatment

In malignant cells, mutations on and dysregulation of assorted cell cycle regulators, such as CDKs, cyclins, CAK, CKI, CDK substrates, and checkpoint proteins, have been frequently observed [2]. It has been well recognized that the expression levels of CDK4/6 are significantly higher in many tumors [6,7,8,9]. Overexpressed CDK4/6 boost G1/S conversion through directly and indirectly (by stimulating CDK2) phosphorylating Rb, facilitating tumorigenesis. In addition to overexpression, CDK4/6 hyperactivation is common in miscellaneous malignancies (breast cancer, lung cancer, prostate cancer, melanoma, leukemia, lymphoma, glioma, sarcoma, etc.) with differentiated tissue preferences for CDK4/6. CDK4 tends to amplify in epithelial tumors (i.e., in varied cancers) and certain sarcomas, while its homolog favors mesenchymal tissues (including leukemias and sarcomas) [4]. Most human tumors retain wild-type Rb [5], and inhibition of overexpressed or hyperactivated CDK4/6 in these cells can arrest the cell cycle in G1. Even in Rb(-) tumors, CDK4/6 inhibitors also function by reducing cell entry into mitosis or inducing apoptosis in an Rb-independent way [10]. Furthermore, targeting CDK4/6 can also inhibit their cell cycle-independent functions in tumorigenesis. Transcriptomic profiling in breast cancer has uncovered that CDK4 modulates inflammatory cytokine signaling [11]. CDK6 can induce angiogenesis, stem cell activation, immune response, etc. [12].

Moreover, many oncogenes cause cancers by activating the CDK4/6-Rb-E2F pathway and inducing cell proliferation. These include but are not limited to JAK/STAT, PI3K/Akt/mTOR, RAS/RAF/MEK/ERK, BTK/NF-ΚB, and Wnt/β-catenin pathways [13]. In addition, mutations in tumor suppressors such as p53 can also activate the CDK4/6-Rb-E2F pathway by releasing p21^CIP1^ inhibition. CDK4/6 thus serves as a hub in tumorigenesis pathways. Especially in ER+ breast cancer, there is mutual activation between ER and cyclin D, which will be clarified further (Figure 2). Knockout studies have shown that CDK4/6 are critical for tumor cell growth whereas may be dispensable in normal cells. It is, therefore, safe to kill the enemy without hurting friendly forces [14]. These features together make CDK4/6 appealing and safe targets for anticancer therapy.

CDKs, their regulators, and substrates are targets of human cancers. Generally, there are two therapeutic interventions: indirectly targeting CDK regulators or directly focusing on CDKs. The complexity of CDK modulation provides diverse possibilities for indirect tactics, involving lessening or raising the quantity of cyclins or CKI severally, changing activity of those regulators, etc. [2]. Direct policies are attainable at different points of gene expression, which primarily embrace clustered regularly interspaced short palindromic repeats system (CRISPR), RNA interference (RNAi), and protein targeted inhibition or degradation technology. In recent years, direct strategy discovery has been a hot area.

## 3. CRISPR and RNAi

Silencing CDKs at DNA or RNA level before protein expression is a fascinating approach to cancer therapy. However, very few growth suppression outcomes using CRISPR or RNAi-mediated CDK4/6 or CDK2 deletion have been detected [11,15,16,17]. Contradictorily, there is evidence denying antineoplastic potency of RNAi-based CDK4/6 or CDK2 knockdown [5,18]. In contrast, CRISPR or RNAi treatment targeting other CDKs (CDK7, -8, -9 for CRISPR and CDK1, -10, -12 for RNAi) [19,20,21,22,23] has successfully diminished tumor proliferation.

The disparity between interphase CDKs (CDK2, -4, -6) and CDK1 regarding responsiveness to CRISPR or RNAi is thought-provoking. Among the four CDKs linked to mitosis, CDK1 alone is enough to drive the cell cycle, while CDK2 and CDK4/6 are only indispensable in cell proliferation of specific cells [4,24]. Meanwhile, the lack of cyclin D-CDK4/6 complex can be compensated by cyclin D-CDK1 or cyclin D-CDK2; similarly, cyclin E-CDK 1 or cyclin E-CDK4 can make up for cyclin E-CDK2 absence [5,25,26]; likewise, CDK1 and CDK2 act in a partially redundant way [27]. Hence, the overlapping jobs of mitogenic CDKs and the relatively dispensable position of CDK2 and CDK4/6 compared with CDK1 may partly account for that discrepancy. Namely, loss of function of CDK1 triggered by CRISPR or RNAi cannot be fully offset by interphase CDKs, while it remains possible for CDK1 to fully compensate inhibition of interphase CDKs.

## 4. Small Molecular Inhibitors

Research on CDK4/6 small molecule inhibitors (SMIs) is an active area. SMIs impede ATP binding to small, determined pockets of CDKs, which holds more potential than hindering large protein-protein interfaces (e.g., interface of cyclin-CDK). However, there are over 500 protein kinases in human genomes, yet ATP binding sites of CDKs are highly conservative [3], so selective drug discovery is challenging. To handle the problem, selectivity determinants outside of the ATP binding pockets are required [28]. The first-generation CDK inhibitors were pan-CDK inhibitors. Due to their poor selectivity and high toxicity, the majority have not been approved for clinical use. The second generation was aimed at improving selectivity for CDK1 and CDK2 and/or total strength [29], and the third generation developed selectivity for CDK4/6 [26]. In addition to poor selectivity and high toxicity, resistance to SMIs cannot be ignored either. Clinical application of CDK4/6 SMIs appears to display weak effects in some tumors (including colorectal cancer, triple-negative breast cancer, melanomas, etc.), where innate or rapid acquisition of resistance may happen [29]. As mentioned, when CDK4/6 are in deficiency, CDK1 or CDK2 can take their roles. Similar events may occur in tumors chronically exposed to CDK4/6 SMIs [5], which may partially explain the mechanisms of resistance, and more reasons will be elucidated hereinafter.

Recent evidence has brought to light diversified mechanisms of tumor suppression of CDK4/6 SMIs apart from cell cycle arrest. These novel paths cover inducing senescence, modulating cell metabolism, enhancing immune response, and so on [1]. Understanding these mechanisms offers opportunities in that the behind principles may explain reasons for drug resistance, thus aiding the development of combination therapy.

### 4.1. One Drug Works Alone

To date, a series of clinical or preclinical evidence has demonstrated effects of CDK4/6 inhibitors in multiple cancers and four drugs, palbociclib (by Pfizer), ribociclib (by Novartis), abemaciclib (by Eli Lilly), and trilaciclib (by G1 Therapeutics), have been approved by Food and Drug Administration (FDA) [30,31] (Figure 3). The first three target HR+/HER2− metastatic breast cancer, and clinical trials for multiple solid tumors of the three drugs are under way [32,33,34,35,36,37]. Trilaciclib is approved to decrease chemotherapy-induced myelosuppression in extensive-stage small-cell lung cancer. Among palbociclib, ribociclib, and abemaciclib, abemaciclib has the most potent efficacy. First, abemaciclib reaches 50% inhibition (IC_50_) of CDK4 and CDK6 at 2 and 5 nM, respectively, while palbociclib needs 9–11 and 15 nM severally to attain the effect, and for ribociclib that reaches 10 and 39 nM [38]. Second, cells adapted to palbociclib show cross-resistance to ribociclib but remain sensitive to abemaciclib [39]. The differences in IC_50_ and resistance among the three drugs are consistent with the special functions of abemaciclib, which can target multiple kinases apart from CDK4/6 (Table 1). In addition, abemaciclib can capture cells in G2 other than G1, which is independent of CDK4/6. Third, abemaciclib is the only CDK4/6 inhibitor drug for the treatment of advanced or metastatic breast cancers that can be used alone.

Besides the above four, multiple CDK4/6 inhibitors have also stepped into clinical trials, such as FCN-437c and XZP-3287, both under phase I clinical trials for advanced solid tumors [40,41]; lerociclib, targeting CDK4/6 and CDK9, under phase I/II clinical trials for ER+/HER2− metastatic breast cancer and EGFR+ metastatic NSCLC [42]; PF-06873600, targeting CDK4/6 and CDK2, under phase I/IIa trials for ER+/HER2− metastatic breast cancer, TNBC and ovarian cancer [43], etc. For more information, readers can refer to some recent references [44,45,46].

### 4.2. Therapeutic Mechanisms beyond G1/S Arrest

The functions of CDK4/6 SMIs were originally believed to induce G1/S arrest and shut down cell proliferation. However, recent studies have revealed that cellular response to SMIs inhibition is much more complex. Arresting cancer cells in the G1 phase with SMIs can also lead to quiescence or senescence. Quiescence is reversible, and quiescent cells can re-enter the cell cycle in response to mitogenic signaling. In contrast, senescence is irreversible and characterized by a senescence-associated secretory phenotype (SASP) [47]. Senescence is also frequently accompanied by changes in chromatin organization, nuclear shape, cytoplasmatic size, cell morphology, or metabolism. Cells exposed to lower doses of CDK4/6 SMIs cause reversible cell cycle arrest, while exposure to higher doses leads to senescence [48,49,50,51]. These observations suggest that targets other than CDK4/6 may be inhibited by high doses of SMIs, and this inhibition is a prerequisite for achieving senescence. This hypothesis is supported by the observation that only higher doses of palbociclib, but not CDK4/6 knockdown, induced senescence [52]. Another model is that the function of CDK4/6 inhibitors is merely to create a senescence permissive state while other intrinsic or external factors determine the cell fate [53]. Senescence-promoting factors include MDM2 reduction, mTOR inhibition, and proteasome hyperactivation [54,55,56,57] (Figure 4).

The defining feature of senescence is the cessation of cell division. Cellular senescence can be triggered by stress factors such as reactive oxidative stress (ROS), irradiation damage, chemotherapeutic agents, oncogenic activation, and anticancer therapies [58]. Senescence, just like apoptosis, has been deemed as a critical anticancer defense mechanism intrinsic to the cell [59]. The other remarkable feature of senescence is SASP, which stimulates the immune clearance of senescent cells [60]. Senescent cells secret SASP factors that attract and activate both innate and adaptive immune systems. Natural killer cells and macrophages are major players in the clearance of senescent cells [61,62]. However, SASP factors can be both beneficial and detrimental. When the immune system cannot clear senescent cells in time, accumulation of these long-lived senescent cells, which overexpress and secret hazardous factors such as inflammatory cytokines, chemokines, and matrix remodeling factors, promotes tumorigenesis [63,64,65,66]. This leads to cancer recurrence that causes even more deleterious effects because therapy-induced senescence reprograms cancer cells into a stem-like state [67]. Obliteration of senescence cells by inducing apoptosis can alleviate these deleterious effects. It has been shown in a mouse model that therapeutic interventions to clear senescent cells improve lifespan [68]. Thus, combined administration with drugs that induce senescence and selectively kill senescent cells (termed senolytics) is an appealing strategy in targeted anticancer therapy [69].

Besides quiescence and senescence, CDK4/6 inhibitors can also induce apoptosis in certain cancer cells. Wang et al., demonstrated that CDK6 can phosphorylate and inhibit two rate-limiting enzymes in the glucose metabolism pathway, PFKP and PKM2, in T-cell acute lymphoblastic leukemia (T-ALL). This forces the glycolytic intermediates into the pentose phosphate and serine pathways, which leads to elevated NADPH and glutathione levels and improved ROS neutralization. Therefore, inhibition of CDK6 by palbociclib results in activation of PFKP and PKM2, depletion of NADPH and glutathione, increase in ROS, and apoptosis of tumor cells [70]. Combined therapies can also induce apoptosis. For example, combination of CDK4/6 inhibitor with MEK inhibitor leads to apoptosis in melanoma cell lines [71]. Combination of palbociclib with PI3K/mTOR inhibitor in pancreatic cancer and combination of CDK4/6 inhibitor with PI3Kα inhibitor in breast cancer also leads to apoptosis [49,72].

The most discouraging outcome of CDK4/6 inhibitor treatment would be drug resistance. Mechanisms of drug resistance and methods to overcome it will be illustrated below.

### 4.3. Drug Resistance Problem

Although CDK4/6 inhibitors have revolutionized treatment for breast cancer, and promising antitumor effects of various inhibitors have also been witnessed in clinical or preclinical trials, resistance to CDK4/6 inhibitors is emerging. In breast cancer, almost all tumors gain resistance to these agents after prolonged therapy. CDK4/6 inhibitors resistance thus has been a big challenge in anticancer therapy. Generally, the reasons contributing to resistance can be roughly grouped into kinase-dependent and kinase-independent approaches. The former includes compensatory activation or overexpression of cyclin D-CDK4/6-Rb axis or other proteins directly or indirectly linked to it, involving other CDKs, cyclins, and assorted oncogenic proteins, transcriptional factors, etc., in different biochemistry pathways. Compensative induction of cyclin D1 was discovered in melanoma patient-derived xenografts (PDXs) [73]. High CDK4 and CDK6 expression were also observed in CDK4/6 inhibitors (CDK4/6i)-resistant glioblastoma multiforme xenograft and ER+ breast cancer cell line models separately [74,75]. ER signaling activation is common in breast cancer, and the crosstalk between cyclin D-CDK4/6 pathway and ER signaling pathways is intriguing. Cyclin D can activate ER by direct or indirect interactions. In turn, ER signaling pathway can increase cyclin D mRNA and protein levels [76,77]. The above evidence may clarify mutual promotion in resistance between CDK4/6i and hormone therapy. Cyclin D-CDK4/6-Rb pathway can also be activated by loss of Rb [78]. On top of that, impaired sensitivity to CDK4/6i may occur when certain CDKs, cyclins or proteins in signaling pathways parallel to or upstream of the axis are triggered, such as CDK7 [79], cyclin E1 [80], YAP and TAZ [81], FGFR [82], PI3K [83,84], c-Met and TrkA-B [85], RAS [86], MDM2 and MDM4 [86,87], CCNE1 and CCNE2 [86,88,89,90], AURKA [86], etc. On the other hand, kinase-independent functions of CDK6, which cannot be targeted by SMIs, also lead to poor response. CDK6 can control transcription of certain genes, thus facilitating angiogenesis and stem cell activation in tumors. For example, in AML and ALL, CDK6 benefits angiogenesis by inducing VEGFA, AKT, FLT3, or aurora kinase (AURK) [91,92,93,94]. In hematologic malignancies, CDK6 can activate hematopoietic stem cells [95], In addition to kinase-dependent and kinase-independent mechanisms of insensitivity to SMIs, CDK4/6i sequestration by increased lysosomal biogenesis may also play a part [96], and more reasons for poor response are to be explored.

### 4.4. Two Drugs Are Better Than One

Resistance to CDK4/6 inhibitors monotherapy calls for more appropriate antitumor strategies, which gives rise to combination therapy and PROteolysis TArgeting Chimera (PROTAC). Details of PROTAC will be discussed later; here, we mainly introduce situations of combination therapy. Understanding mechanisms of resistance offer opportunities in developing combination therapy [29]. As mentioned above, in the process of resistance, multiple proteins besides CDK4/6 may overexpress or be hyperactivated. Therefore, one possible way to counteract resistance is focusing on two or more targets at the same time. Synergistic tumor-suppressing effects of CDK4/6 inhibitors and other selective inhibitors have been described in numerous clinical or preclinical studies, including CDK4/6i in combination with EGFR [97], PI3K [98] or FGFR inhibitors [99] in breast cancer xenografts, IGF1 inhibitors in pancreatic adenocarcinoma [100], MEK inhibitors in NRAS mutant melanoma [101], inhibitors of other targets such as MET/ERK [85], HER2 [102], MDM2/4 [73,103], eukaryotic initiation factor [104] and even pan-CDK inhibitors [105], etc. in various cancers. Besides targeting the above kinases, hormone therapy aimed at estrogen or ER combined with palbociclib, ribociclib, or abemaciclib has become a standard treatment in metastatic ER+/HER2− breast cancer. Both palbociclib and ribociclib received approval for treating ER+/HER2− postmenopausal women in combination with aromatase inhibitors [106,107], which prevent estrogen formation from androgen, while abemaciclib was approved in combination with ER antagonist fulvestrant [108]. Among the three drugs, only abemaciclib can be used as a monotherapy for ER+/HER2− breast cancer [34], which may be due to its special functions mentioned earlier.

Furthermore, CDK4/6i combined with immune checkpoints inhibitors is worth considering. Zhang et al., demonstrated a negative correlation between PD-L1 expression and CDK4 activity [109]. Therefore, CDK4/6 inhibitors may induce an increased PD-L1 level, which can bring out both pros and cons. On the one hand, elevated PD-L1 expression facilitates immune evasion of tumor cells. On the other hand, this may increase sensitivity to PD-L1 blockade [109]. Correspondingly, outcomes of immune combination therapy have been complex. Although abemaciclib [110], and palbociclib [111], combined with PD-L1 blockade, have both been observed enhancing therapeutic efficacy in vitro and exhibited no unexpected safety profiles in several clinical trials [111,112], greater toxicity compared with monotherapy was also reported in stage IV NSCLC [113]. Hence, further assessment of the combination is needed.

Apart from targeted therapy and immune therapy, chemotherapy and radiotherapy are also major treatments in anticancer strategies. Classical chemotherapy targets cells in the S phase of mitosis, while CDK4/6i arrests cells in the G1 phase. Therefore, CDK4/6i has long been thought to antagonize the effects of cytotoxic agents (DNA- or mitosis-damaging chemotherapeutic drugs, such as cisplatin, gemcitabine, etc.), and there are several facts [114,115,116]. However, recent evidence suggests that CDK4/6i administration after chemotherapy protects cells from toxicity induced by the latter [117,118,119], and similar cell-protection effects under sequential administration also apply to radiotherapy [120,121]. In particular, CDK4/6 inhibition after chemotherapy or radiotherapy can also enhance tumor suppression in some settings [122,123,124].

Combination therapy is encouraging in prevailing over resistance and promoting antitumor effects (Table 2). Some success in this realm has been witnessed, and more research and assessments are still needed.

### 4.5. Kill Two Birds with One Stone

Combination therapy may help delay or overcome drug resistance and meanwhile exert synergistic anticancer effects, one of the approaches of which is targeting additional targets besides the cyclin D-CDK4/6-Rb pathway. Therefore, drugs that intrinsically inhibit two or more targets at the same time are more powerful cancer-killers than those single-targeted ones. Reddy et al., reported a compound that could inhibit CDK4/6, ARK5, PI3K-δ, PDGFR-β, and FGFR1 simultaneously, which are all linked to tumorigenesis, survival, and metastasis [125]. A CDK4 inhibitor found by Novartis was also a multi-kinase inhibitor, showing activity toward 13 kinases, including PKA, ALK, JAK1, etc. [126]. In addition, AMG 925 by Amgen [127], a dual-kinase inhibitor aimed at CDK4 and fms-like tyrosine kinase (FLT) 3, may help overcome FLT3 resistance in AML [128]. Another compound synthesized by Wang et al., also exhibited inhibitory activities on CDK4/6 and FLT3 and is now in phase I clinical trials [129].

Discovery of dual- or multi-kinase inhibitors is serendipitous, and more effective remedies are waiting to be developed.

## 5. PROTACs

PROteolysis TArgeting Chimera (PROTAC) is a game-changing technology and has the potential to revolutionize drug discovery [130]. PROTACs are bifunctional molecules that, instead of inhibiting activity, destroy protein of interest (POI) via hijacking the ubiquitin proteasome system (UPS). The two ligand moieties of PROTAC that bind to E3 ligase and POI, respectively, are connected by a linker of different lengths and compositions. This design brings POI in the vicinity of E3 ligase, promoting the formation of new interactions between POI and E3 ligase that do not exist in physiological conditions. The formation of POI-PROTAC-E3 ligase ternary complex facilitates the transfer of ubiquitin from E2 ubiquitin-conjugating enzymes to surface-exposed lysine residues in POI where ubiquitination occurs. Ubiquitin can be attached end-to-end to a single lysine residue, which is called poly-ubiquitination. The modified POIs are then recognized by proteasome for degradation, sparing PROTACs for the next cycle of degradation. Selection of E3 ligase is critical for PROTAC design, and there are several principles: (1), Oncogenic E3 is better than tumor suppressor E3; (2), E3 only expressed or highly expressed in specific cancer types; (3), Small molecule ligand (and preferably the structure of E3-ligand complex) is available. The most frequently used E3 ligases in PROTAC design include CRBN, VHL, MDM2, and cIAP2 (Figure 5). Compared to the classical SMIs, PROTACs have several advantages. First, SMIs only inhibit POI with enzymatic activity, while PROTACs destroy protein of any type, given that it binds a ligand. Second, drug resistance is very common in SMIs but very rare for PROTACs. Third, PROTACs exhibit better specificity compared to the corresponding SMIs [131].

These superiorities of PROTAC have attracted great attention both from academia and industry for developing new types of therapeutic agents, in particular anticancer drugs [132]. So far, more than one hundred target proteins have been reported to be degraded successfully by this technology. More than 10 PROTACs have been further developed and are in clinical trials for use in various cancer types. ARV-110 targeting androgen receptor (AR) and ARV-471 targeting estrogen receptor (ER), both of which are developed by Arvinas Inc. (New Haven, CT, USA) to combat prostate cancer and breast cancer, respectively, came into clinical trials in 2019 [132]. Soon afterward, several degraders entered clinical trials: ARV-766 (Arvinas) and AR-LDD (Bristol Myers Squibb) targeting AR for prostate cancer; DT2216 (Dialectic) targeting BCL-XL for liquid and solid cancers; KT-413 (Kymera) targeting IRAK4 for diffuse large B cell lymphoma (DLBCL); KT-333 (Kymera) targeting STAT3 for liquid and solid cancers; NX-2127 and NX-5948 (Nurix) targeting BTK for B cell malignancies; CG001419 (Cullgen) targeting tropomyosin receptor kinase (TRK) for cancer; CFT8634 (C4 Therapeutics) and FHD-609 (Foghorn) targeting BRD9 for synovial sarcoma. Just some of them are listed here.

The resistance to CDK4/6 inhibitors makes PROTAC technology an attractive alternative strategy [133]. Drug-resistant mutated forms of CDK4/6 implicated in cancers could also be destroyed using this technique. In addition, the enhanced specificity of PROTACs brings about less toxicity. Another reason for choosing PROTAC technology is its ability to anchor the non-kinase activity. It is reported that in addition to enhancing proliferation, CDK6 can also stimulate angiogenesis independent of its kinase activity in hematopoietic malignancies [91]. Such a role is beyond the reach of current CDK4/6 inhibitors. Therefore, more and more researches focus on CDK4/6 drug development using PROTAC technology [134].

### 5.1. PROTACs Targeting Both CDK4/6

It is required to silence both CDK4 and CDK6 to achieve G1/S arrest due to their redundant roles in the cell cycle. Actually, most PROTACs targeting CDK4/6 can destroy both of the kinases due to their similar structural folding and common ligands. Zhao et al., reported the first potent CDK4/6 degrader in 2019. The **Degrader 1** was made by connecting pomalidomide, a ligand of E3 ligase CRBN, and also a drug to treat certain types of cancer such as multiple myeloma and Kaposi sarcoma, to palbociclib (Figure 6). Treatment of MDAMB-231 cells, a triple-negative (estrogen receptor, progesterone receptor, and HER2-negative) breast cancer cell line, with the degrader results in the efficient degradation of CDK4 and CDK6 with DC_50_ (the concentration at which 50% of the target protein has been degraded) values of 13 and 34 nmol/L, respectively, which subsequently decreases the level of the Rb phosphorylation in a dose-dependent manner [135].

### 5.2. PROTACs Targeting CDK4 or CDK6 Alone

Although the sequence and function of CDK4 and CDK6 are highly conserved, homolog-specific functions have been described. It was reported that conditional ablation of *Cdk4*, but not *Cdk6*, induces an immediate senescence response only in lung cancer cells expressing an endogenous K-Ras oncogene [136]. In contrast to CDK4, many CDK6-specific functions have been reported. Kollmann et al., reported that CDK6 regulates the transcription and expression of VEGF-A, a protein involved in angiogenesis, and the tumor suppressor p16INK4a, independent of its kinase activity [91]. CDK6 can also interact with NFκB to regulate the expression of specific inflammatory genes [137]. Deng et al., showed that CDK6, but not CDK4, activates T cells via phosphorylating NFAT family proteins, which results in improved antitumor immune response [62]. Interestingly, Wang et al., demonstrated that two rate-limiting enzymes in the glucose metabolism pathway in tumor cells are substrates of CDK6 but not CDK4. Phosphorylation of these two enzymes eventually results in elevated NADPH and glutathione levels and improved ROS neutralization, by which tumor cells evade apoptosis [70].

The homolog-specific functions, especially those in cancer cells, make the CDK4- and CDK6-specific therapeutic agents highly desirable. Moreover, acquired CDK6 amplification after prolonged exposure to CDK4/6 inhibitors promotes drug resistance, potent and selective inhibition or degradation of CDK6 may potentially overcome drug resistance [74]. However, current SMIs drugs target the ATP-pockets while the residue composition of the pockets in CDK4 and CDK6 share 94% sequence identity; thus, these drugs can not target CDK4 or CDK6 exclusively.

Selective degradation by PROTACs requires the formation of high affinity, long-lasting, differential E3 ubiquitin ligase–PROTAC-POI ternary complex [138,139]. PROTACs bring POI in the vicinity of E3 ligase, promoting the formation of new interactions between POI and E3 ligase that otherwise do not exist in normal conditions. Thus, in addition to the ligands, the amino acid differences in the POI-E3 interface determine the specificity of degradation. Albeit highly conserved in their ATP binding pocket, sequence alignment shows that surface-exposed residues of CDK4 and CDK6 share 51% identity. Another factor affecting PROTAC degradation is the distribution of surface-exposed lysine residues in POI, which are the substrates of ubiquitination by E3 ligase. However, both the number and distribution of surface-exposed and ubiquitin ligase-accessible lysine of CDK4 and CDK6 are quite different [140]. These disparities between them thus make the CDK4- and CDK6-specific PROTACs attainable.

Brand et al., achieved CDK6-specific degradation by designing a **Degrader 2**, which also links palbociclib to pomalidomide with a linker. **Degrader 2** degrades CDK6 in a time and concentration-dependent manner. Global proteomics result shows that only CDK6 is destroyed while other proteins, including CDK4, remain unchanged. **Degrader 2** decreases the phosphorylation level of Rb and causes cell cycle arrest to inhibit cell proliferation in CDK6-dependent AML cell lines [141]. Another CRBN-based and palbociclib-derived PROTAC molecule was described by Rana et al., **Degrader 3,** with a very similar structure to **Degrader 2**, specifically and efficiently degraded CDK6 while sparing CDK4 in HPNE and MiaPaCa2 cell lines [140].

To overcome the limitations of CRBN-based PROTACs, Steinebach et al., systematically tested different E3 ligases and their respective ligands connecting to palbociclib with various linkers. They finally identified a VHL-based PROTAC (**Degrader 4**) that can degrade CDK6 specifically [142].

These CRBN-based and palbociclib-derived PROTACs were demonstrated to selectively degrade CDK6 over CDK4 even though palbociclib itself has almost equal inhibitory potency and affinity against both kinases. The preferred degradation of CDK6 over CDK4 was confirmed by a recent study [143] and could be explained by the number of surface-exposed and ubiquitin ligase-accessible lysine residues. There are 16 such lysine residues in CDK6 while only 9 in CDK4, which makes CDK6 more susceptible to ubiquitination and subsequent degradation (Figure 7). The other factor may be that the surface structure, especially that in close proximity to palbociclib pocket, of CDK6 matches better to that of CRBN than that of CDK4 does [144].

### 5.3. PROTACs Targeting Multiple CDKs

CDK4 or CDK2 can phosphorylate SMAD3, a transcription factor mediating the transforming growth factor-β (TGF-β) antiproliferative response. Phosphorylation of SMAD3 blocks its transcription and antiproliferative activity; this, in turn, decreases the level of p15, a member of the INK4 inhibitor family, accordingly promoting the G1/S transition. Therefore, simultaneous inhibition of CDK4/6 and CDK2 leads to cell cycle arrest induced by TGF-β [145]. Breast cancer cells treated with palbociclib have shown drug resistance because CDK2 takes the roll of CDK4. However, the Brk-SH3 peptide, ALT, inhibits kinase activities of both CDK2 and CDK4 indirectly, leading to potent and prolonged cell cycle arrest. Accordingly, drugs targeting CDK2, CDK4/6 may exhibit effective treatment and surmount the resistance to the CDK4/6 inhibitor [105].

Recently, Wei et al., developed a PROTAC degrader (**Degrader 5**) based on a ribociclib derivative and a CRBN ligand. **Degrader 5** destroys CDK2/4/6 simultaneously and causes cell cycle arrest and apoptosis of melanoma cells, suggesting that PROTACs targeting CDK2/4/6 present a promising approach for treatment of cancers. The authors further improved the oral bioavailability of the compound by adding a lipophilic group at the active site of CRBN ligands. P.O administration of the improved PROTAC significantly suppressed B16F10 tumor growth in mice [146].

### 5.4. PROTACs Targeting CDK4/6 as Well as Other Targets

Most PROTACs degrade only one target protein. Some destroy their highly related homologs as well. However, in certain instances, PROTACs can also degrade other targets because the ligand employed exhibits affinity toward these proteins as well. Jiang et al., showed such cases. They designed dual degraders of CDK4/6 (**Degrader 6**, **Degrader 7**, **Degrader 8**) which showed potent degradation of Ikaros (IKZF1) and Aiolos (IKZF3), well-established “off-targets” of imide-based degraders. Surprisingly, they found that combined degradation of IKZF1/3 and CDK4/6 displayed synergic antiproliferative effects in lymphoma cell lines [28].

## 6. Conclusions and Perspectives

According to World Health Organization (WHO) statistics, cancer remains one of the most serious threats to human health worldwide. It ranks as the second leading cause of death. Cancers are characterized by abnormal and uncontrollable cell division. Hyperactivation and overexpression of CDKs are often the drivers of cancer pathogenesis. Accordingly, targeting CDKs has become the preferred strategy to combat cancer [147]. Among their relative homologs, CDK4/6 are critically important as they regulate the phosphorylation state of Rb both directly and indirectly. It is well acknowledged that the vast majority of human cancers show dysregulation of the CDK4/6-Rb-E2F pathway via very diverse mechanisms, and the cyclin D-CDK4/6 complex is hyperactivated in many types of human cancers. It is therefore estimated optimistically that the vast majority of human cancers can be treated with drugs targeting CDK4/6, either alone or in combination with other drugs.

In the past decade, great achievements have been made in the field of anticancer treatment, especially drugs targeting CDK4/6. The approval of palbociclib as a treatment (in combination with letrozole) for patients with estrogen receptor-positive advanced breast cancer by the FDA in 2015 is a milestone in the field. Since that, a few me-too drugs came into clinical trials, with ribociclib, abemaciclib, and trilaciclib being approved for cancer treatment subsequently. These inhibitors have been used in clinics to treat various cancers, including, but not limited to, breast cancer, non-small-cell lung cancer, prostate cancer, and acute myeloid leukemia [29,148,149].

It is obvious to think that CDK4/6 inhibitors combat cancer by causing cell cycle arrest, thereby inhibiting cell proliferation. However, CDK4/6 SMIs can do more than that. Recently, it has been reported that treatment with CDK4/6 SMIs leads to quiescence, senescence, or apoptosis. The outcomes of SMIs inhibition beyond cell cycle arrest could be, at least partially, explained by the “off-target” effects. Indeed, all three approved SMIs can target kinases other than CDK4/6, which is not surprising considering their similar ATP binding pockets (Table 1). The cellular fate is determined by intrinsic or external factors, such as cell type, dysregulation of specific proteins, and different inhibitor drugs. Nevertheless, the decisive factor of cellular fate remains elusive. To uncover the myth, more efforts are needed to discover unknown substrates of CDK4/6 and unknown “off-targets” of CDK4/6 SMIs.

Aside from their side effects, the biggest challenge and the most desperate outcome of CDK4/6 inhibition is drug resistance. Mechanisms for drug resistance include kinase-dependent and kinase-independent approaches. The former involves compensatory activation or overexpression of cyclin D-CDK4/6-Rb axis or other related proteins, and kinase-independent function of CDK4/6 cannot be targeted by SMIs, resulting in a poor response. Several strategies have been applied to overcome drug resistance. Combination therapy is a cornerstone of cancer therapy. When combined use with other drugs, CDK4/6 inhibitors exhibit a synergistic effect, leading to improved efficacy. Tumor compensation is the underlying mechanism for most drug resistances. By blocking more than two pathways at once, the tumor is less likely to compensate. Sometimes combination therapy can be realized by one drug, given that it targets two pathways simultaneously. PROTAC technology is another way to combat drug resistance [133]. Unlike traditional SMIs, which only silence the partial function of target protein, PROTACs destroy the whole molecule. Thus drug-resistant mutants can also be degraded by PROTACs. In addition, drug resistance arising from CDK6 overexpression can be tackled by CDK6-specific PROTAC degradation. Some PROTACs target proteins in two pathways involved in tumorigenesis, where they function in a way similar to combination therapy. So far, all PROTACs aiming to degrade CDK4/6 are based on the first three approved inhibitors (palbociclib, ribociclib, and abemaciclib). The potential benefit of such design is that PROTACs can inherit the properties of the inhibitors, i.e., to inhibit (or even degrade) “off-targets” of CDK4/6. The unexpected degradation of targets other than CDK4/6 might also be detrimental. It is therefore critical to evaluate the toxicity effect of PROTACs with such design.

PROTACs open a new chapter for novel anticancer drug development, and more than a dozen of PROTAC drugs have come into clinical trials in the past three years. Albeit great success in academia and representation of next-generation drugs, PROTACs have their limitations. First, so far, there is no structure of CDK4/6-PROTAC-E3 ligase ternary complex available to uncover the detailed mechanism and guide rational design. Second, PROTACs designs are restricted by the limited number of E3 ligase ligands. Third, PROTACs can induce cellular toxicity caused by non-specific degradation. Fourth, PROTACs usually have poor cellular permeability compared with their corresponding SMIs due to the larger molecular weight.

Although great achievements have been made in the field, further studies are needed to disclose the “off-targets” of CDK4/6 SMIs and link the targets with outcomes of SMIs treatments. The detailed mechanism of SMIs treatment will promote the development of specific drugs with less toxicity and weaker side effects. It will also facilitate the design of combination therapy and PROTACs. PROTACs represent the next generation of drugs, and with great efforts being made to overcome the limitations, we expect that more potent PROTAC drugs with less toxicity will come into clinical trials in the near future.

## Figures and Tables

**Figure 1 biomedicines-10-00685-f001:**
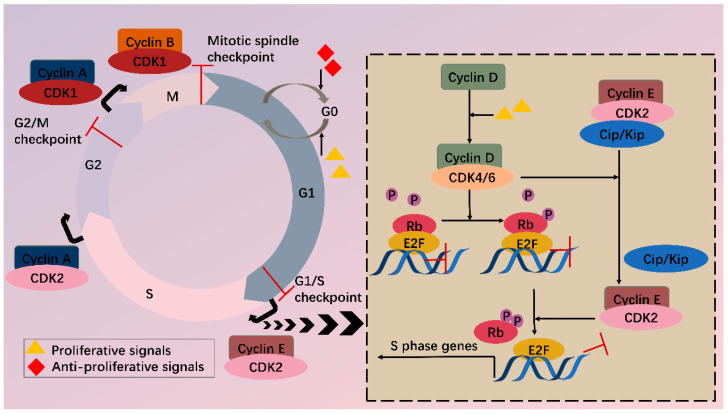
CDKs and cell cycle. Cell cycle consists of G1, S, G2 phase and mitosis. Independent of a cell cycle also exists a dormant G0 phase. Whether a cell steps into a cell cycle relies on balance between proliferative and antiproliferative signals. During a cell cycle, multiple CDKs are sequentially activated. In G1, activated CDK4/6 by cyclin D phosphorylate Rb, partially relieving inhibition of E2F by Rb. Meanwhile, CDK4/6 hijacks Cip/Kip proteins, which stimulate CDK2-cyclin E, facilitating complete phosphorylation of Rb. Thus, E2F activity is totally released, and transcription of S phase genes is initiated. In late G1, CDK2-cyclin E complex is formed, driving transition toward the S phase. Next, CDK2 and CDK1 are successively activated by cyclin A and contribute to S/G2 and G2/M conversion, respectively. Finally, CDK1-cyclin B complex functions during mitosis. Besides CDKs, checkpoints also participate in the cell cycle via regulation of CDKs activity, inducing cell cycle arrest when abnormal cell division is detected.

**Figure 2 biomedicines-10-00685-f002:**
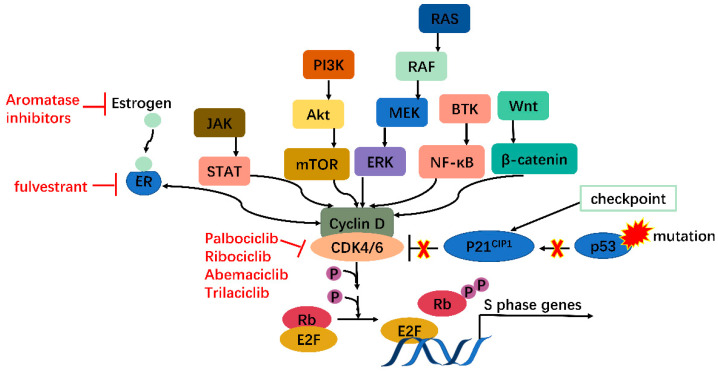
CDK4/6 serves as a hub in tumorigenesis. In cancer, multiple oncogenes may be activated, including those on JAK/STAT, PI3K/Akt/mTOR, RAS/RAF/MEK/ERK, BTK/NF-κB, Wnt/β-catenin pathways, etc., all of which meet at CDK4/6-cyclin D complex. Moreover, mutations on tumor suppressor genes such as p53 can enhance CDK4/6 activity via releasing P21^CIP1^ inhibition. CDK4/6, therefore, serves as a hub in tumorigenesis.

**Figure 3 biomedicines-10-00685-f003:**
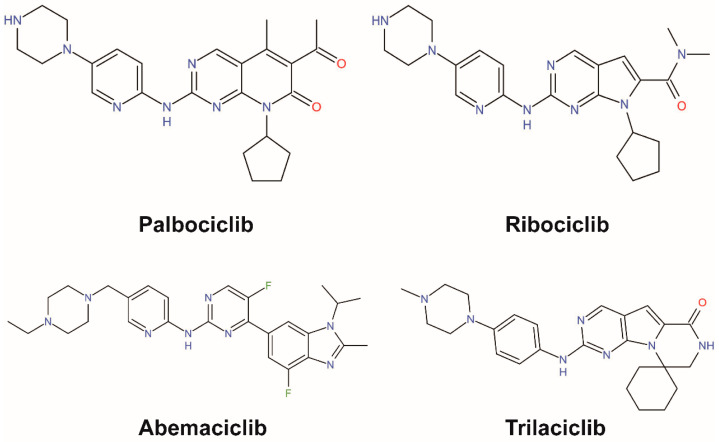
Chemical structures of CDK4/6-specific small molecular inhibitors approved by the FDA.

**Figure 4 biomedicines-10-00685-f004:**
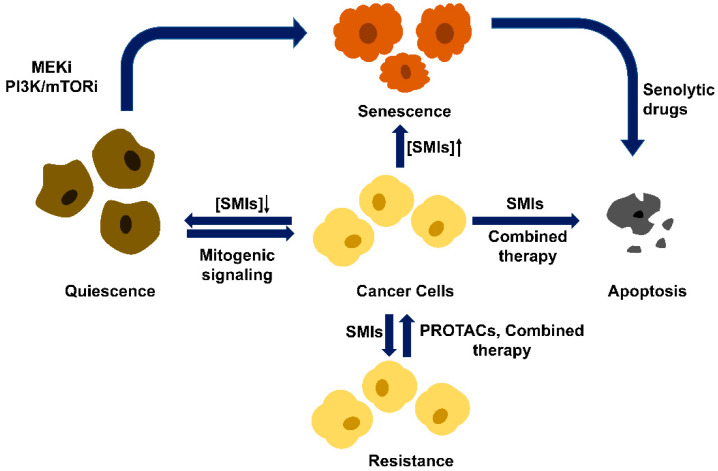
Different cell fates of CDK4/6 small molecular inhibitor (SMIs) treatment. Cancer cells arrested in the G1 phase with SMIs lead to quiescence (at low concentration), senescence (at high concentration), or, in some cancer types, apoptosis. Quiescence is reversible, and quiescent cells can re-enter the cell cycle in response to mitogenic signaling. Senescence is irreversible, and senescent cells can be killed by senolytic drugs and step into apoptosis. Thus, cancer cells can be cleared in two sequential steps. In some cancer types, SMIs alone or the combined treatment with MEK inhibitor or PI3K/mTOR inhibitor can change cell fate toward senescence or apoptosis. Drug resistance is another outcome of SMIs treatment and can be surmounted by combined therapy and PROTACs.

**Figure 5 biomedicines-10-00685-f005:**
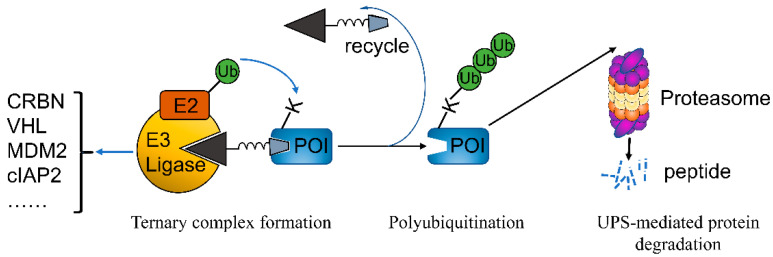
Mode of protein degradation using PROTACs. PROTACs bring the protein of interest (POI) in the vicinity of E3 ligase (such as CRNB, VHL, MDM2, cIAP2, etc.), forming a stable ternary complex of POI-PROTAC-E3 ligase. This structure facilitates ubiquitin transfer from E2 conjugating enzyme to a lysine residue in POI, resulting in ubiquitination of POI. Ubiquitination can occur on ubiquitin itself, leading to polyubiquitination. Polyubiquitinated POI is then subjected to proteasome for degradation via ubiquitin proteasome system (UPS). PROTAC degrader can be recycled for the next round of degradation.

**Figure 6 biomedicines-10-00685-f006:**
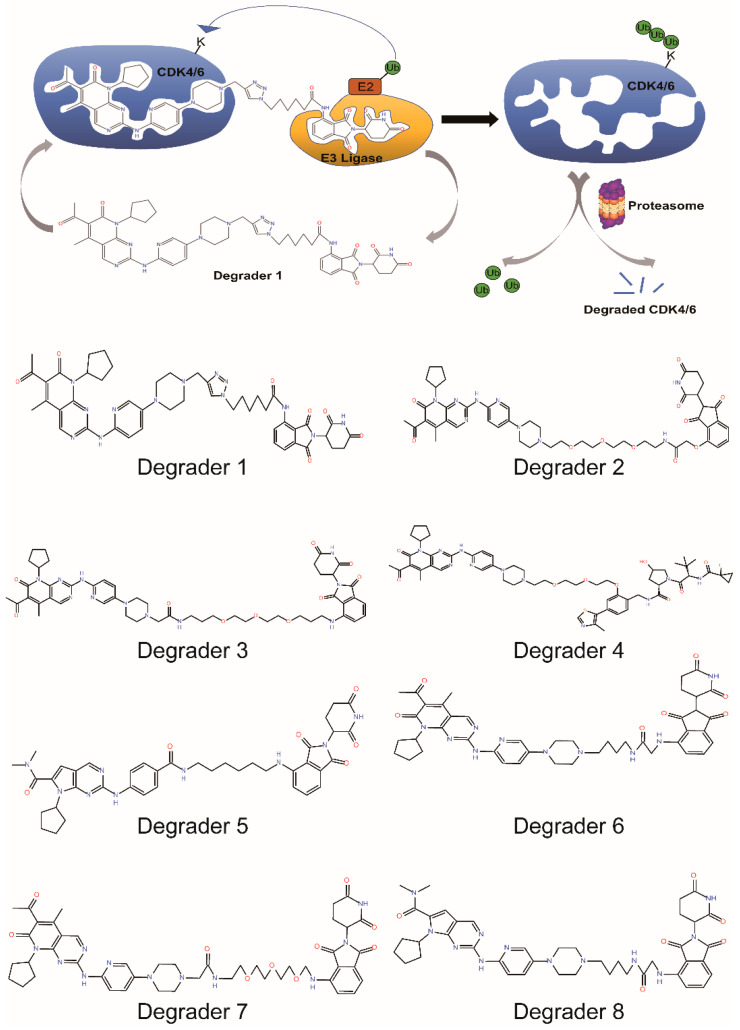
Mechanism of action of PROTACs in the degradation of CDK4/6 and chemical structures of selective CDK4/6 degraders.

**Figure 7 biomedicines-10-00685-f007:**
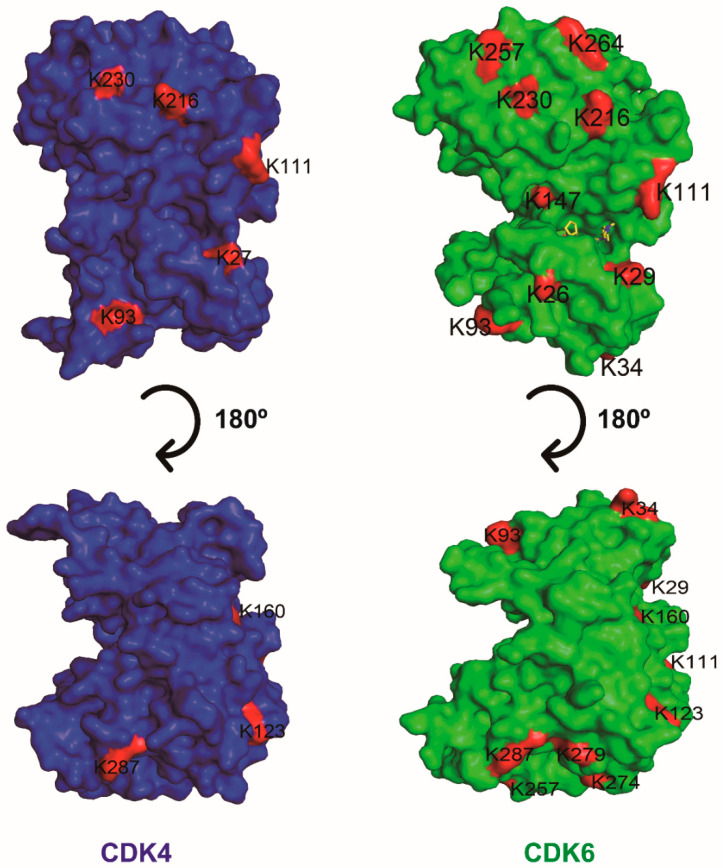
Surface drawing of the human CDK4 (blue) and CDK6 (green) structures in two different orientations. Surface-exposed lysine residues are colored red and labeled.

**Table 1 biomedicines-10-00685-t001:** Comparison of CDK4/6 inhibitors.

Drug	Palbociclib (Pfizer) (PD0332991, lbrance)	Ribociclib (Novartis) (LEE011)	Abemaciclib (Eli Lilly) (LY2835219)
IC_50_ (in vitro kinase assay, recombinant proteins)	CDK4: 9-11 nM CDK6: 15 nM	CDK4: 10 nM CDK6: 39 nM	CDK4: 2 nM CDK6: 5 nM CDK9: 57 nM
Cell cycle arrest	G1	G1	G1, G2
Off-targets	TTK, TRK, ULK2-3, PIK3R4, PIP5K2C, PIK3CD, PIP4K2A/C CAMK2δ, FLT3, PIP4K2B, HIPK1-3, MPSK1, CDK16/17, CLK1-2, CSNK2α	TTK, GAK, QIK, CAMK2δ, CAMK2α-γ	CDK1/2/5/7-9/12-17/19, CAMK2δ, CAMK2α-γ, FLT3, PIP4K2B, HIPK1-3, MPSK1, CLK1-2, CSNK2α1-2, ERK8, GSK3α/β, PIM1/3, AAK1, IRAK1, DYRK1-3, CSNK2α3

**Table 2 biomedicines-10-00685-t002:** Representative clinical trials using CDK4/6i in combination therapy.

CDK4/6i	Cancer Type	Combination Therapy	Trial Name
Palbociclib	Mantle cell lymphoma	ibrutinib	NCT03478514
	Metastatic colorectal cancer	cetuximab	NCT03446157
	Multiple myeloma	bortezomib; dexamethasone	NCT00555906
	Ewing sarcoma	ganitumab	NCT04129151
	Advanced KRAS mutant non-small cell lung cancer	binimetinib	NCT03170206
	Acute myeloid leukemia	vyxeos	NCT03844997
	Squamous cell carcinoma of the head and neck	carboplatin	NCT03194373
	Breast cancer	Letrozole	NCT01740427 (PALOMA-2)
	Breast cancer	fulvestrant	NCT01942135(PALOMA-3)
	Breast cancer	anastrozole	NCT01723774
	Breast cancer	paclitaxel	NCT01320592
Ribociclib	Recurrent platinum sensitive ovarian cancer	paclitaxel; carboplatin	NCT03056833
	ALK-positive non-small cell lung cancer	ceritinib	NCT02292550
	Liposarcoma	siremadlin	NCT02343172
	Metastatic pancreatic adenocarcinoma	everolimus	NCT02985125
	Triple-negative breast cancer	bicalutamide	NCT03090165
	Squamous cell carcinoma of the head and neck	cetuximab	NCT02429089
	Acute lymphoblastic leukemia	everolimus; dexamethasone	NCT03740334
	Prostate cancer	enzalutamide	NCT02555189
	Breast cancer	letrozole	NCT01958021
	Breast cancer	fulvestrant	NCT02422615 (MONALEESA-3)
	Breast cancer	tamoxifen, NSAI	NCT2278120 (MONALEESA-7)
Abemaciclib	Renal cell carcinoma metastatic	sunitinib	NCT03905889
	Recurrent glioblastoma	bevacizumab	NCT04074785
	Hepatocellular carcinoma	nivolumab	NCT03781960
	Non-small cell lung cancer	necitumumab	NCT02411591
	Gastroesophageal cancer Adenocarcinoma	pembrolizumab	NCT03997448
	Non-small cell lung cancer	erlotinib	NCT02152631
	Brain metastases	GDC-0084 entrectinib	NCT03994796
	Breast cancer	anastrozole	NCT02441946 (NeoMONARCH)
	Breast cancer	fulvestrant	NCT02107703 (MONARCH-2)
Trilaciclib	Metastatic triple-negative breast cancer	gemcitabine plus carbo-platin	NCT02978716
	Extensive-stage small cell lung cancer	etoposide, carboplatin	NCT02499770
	Extensive-stage small cell lung cancer	etoposide, carboplatin, atezolizumab	NCT03041311
	Extensive-stage small cell lung cancer	topotecan	NCT02514447

## Data Availability

Not applicable.

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
