# Peer review of "Targeting CDK4/6 for Anticancer Therapy"

_biomedicines, 2022, doi:10.3390/biomedicines10030685_

Round 1

Reviewer 1 Report

The authors in this review summarize the current status in targeting CDK4/6 pathway, describing the approaches to overcome drug resistance, including combination therapy and PROTACs drugs. The work is well written but several issues need to be addressed to consider it suitable for publication.

  • In order to improve the clinical relevance of the review, a table in section 4.1 should be added rather than the drug chemical structures, indicating the CDK4/6i approved in the clinic, in which cancer type, oncologic setting (i.e. adjuvant, neoadjuvant, metastatic), combination therapy, date of approval, etc….
  • In the section 4.3 other important works and concepts for drug resistance mechanisms should be added i.e. McCartney A, et al. Front Oncol. 2019. and Palleschi M, et al. Diagnostics (Basel). 2020
  • Figures must be improved. The quality is really bad, especially for the text.

In general, a more translational tone towards the clinical world is needed.

Author Response

Review 1:

The authors in this review summarize the current status in targeting CDK4/6 pathway, describing the approaches to overcome drug resistance, including combination therapy and PROTACs drugs. The work is well written but several issues need to be addressed to consider it suitable for publication.

Response: We thank the reviewer for the positive comments.

  1. In order to improve the clinical relevance of the review, a table in section 4.1 should be added rather than the drug chemical structures, indicating the CDK4/6i approved in the clinic, in which cancer type, oncologic setting (i.e. adjuvant, neoadjuvant, metastatic), combination therapy, date of approval, etc….

Response: We thank the reviewer for this really great suggestion. We have added a table (table 2) with cancer type, combination therapy and date of approval (trial name) information and we believe that this information will definitely improve readability of the manuscript.

  1. In the section 4.3 other important works and concepts for drug resistance mechanisms should be added i.e. McCartney A, et al. Front Oncol. 2019. and Palleschi M, et al. Diagnostics (Basel). 2020

Response: We have added these two references (77 and 84 respectively) in the revised manuscript.

  1. Figures must be improved. The quality is really bad, especially for the text.

Response: Figures incorporated in the manuscript may not be of high quality enough. However, we uploaded the high-resolution figures along with the manuscript.

Reviewer 2 Report

The review described by Qi and Ouyang summarizes CDK4/6 and the potential utilization of CDK4/6 inhibition in human cancers. While there are many reviews describing CDK4/6 in the context of cancer, cell cycle, therapy and therapy resistance, this review is summarizing CDK4/6 comprehensively. The functions, regulations, and therapeutic intervention regarding CDK4/6 were already very well described in other reviews and this review added very little in this field. However, the authors added a relatively new concept of CDK4/6 inhibition using PROTAC, which I think should be expanded and described more clearly to attract more readers. The suggestions where the authors should improve are described as follows.

  1. Figure 5 is a very simple model that can be found everywhere. Adding more information such as CRBN, VHL in this figure might be helpful to understand more about the concept of PROTAC.
  2. Since the review focuses on more PROTAC, it would be helpful if the authors could add more information of the degradation of CDK4/6 and pomolidomide in section 5.
  3. Showing only the structure of many degraders in Fig 6 is not informative. Adding mechanisms of action wherein the degrader works would attract more readers to go through the figures.

Author Response

Reviewer 2:

The review described by Qi and Ouyang summarizes CDK4/6 and the potential utilization of CDK4/6 inhibition in human cancers. While there are many reviews describing CDK4/6 in the context of cancer, cell cycle, therapy and therapy resistance, this review is summarizing CDK4/6 comprehensively. The functions, regulations, and therapeutic intervention regarding CDK4/6 were already very well described in other reviews and this review added very little in this field. However, the authors added a relatively new concept of CDK4/6 inhibition using PROTAC, which I think should be expanded and described more clearly to attract more readers. The suggestions where the authors should improve are described as follows.

Response: We thank the reviewer for the overall positive comments. We have addressed this issue with more detailed mechanism of how PROTAC works, how to choose E3 for PROTAC design in section 5 in the revised manuscript (see page 9, line 359-371).

  1. Figure 5 is a very simple model that can be found everywhere. Adding more information such as CRBN, VHL in this figure might be helpful to understand more about the concept of PROTAC.

Response: We thank the reviewer for this great suggestion and we have modified Figure 5 by adding more E3 information along with the figure legends.

  1. Since the review focuses on more PROTAC, it would be helpful if the authors could add more information of the degradation of CDK4/6 and pomolidomide in section 5.

Response: These are excellent suggestions. We have added some information in section 5 and 5.1.

  1. Showing only the structure of many degraders in Fig 6 is not informative. Adding mechanisms of action wherein the degrader works would attract more readers to go through the figures.

Response: This is a great point. We have modified Fig 6 with additional cartoon drawing of mechanism of action and corrected figure legends correspondingly.

Round 2

Reviewer 1 Report

I appreciate the efforce of the authors in answering the queries. The manuscript has been sufficiently improved but there are still few issues:

  • many typos are present, please correct.
  • Table 1 is reapeated in two different sections, please correct.
  • both tables are displayed as images and the quality is bad. Please insert them as text. 

Author Response

Review 1:

I appreciate the efforce of the authors in answering the queries. The manuscript has been sufficiently improved but there are still few issues:

Response: We thank the reviewer for the suggestions.

  1. many typos are present, please correct.

Response: We thank the reviewer for this suggestion. We have checked our manuscript under the proofreading mode and made several corrections.

2.Table 1 is reapeated in two different sections, please correct.

Response: We have deleted table 1 in section 5 in the revised manuscript.

  1. both tables are displayed as images and the quality is bad. Please insert them as text.

Response: We have uploaded both tables as images of high quality separately.

Reviewer 2 Report

The authors improved their manuscript by adding explanations based on the reviewer's suggestions.

Author Response

Reviewer 2:

The authors improved their manuscript by adding explanations based on the reviewer's suggestions.

Response: We thank the reviewer for the comments.

Reviewer 2:

The authors improved their manuscript by adding explanations based on the reviewer's suggestions.

Response: We thank the reviewer for the comments.
